

# Comparative demography of commercially-harvested snappers and an emperor from American Samoa

Brett M. Taylor[1], Zack S. Oyafuso[2], Cassandra B. Pardee[1],
Domingo Ochavillo[3] and Stephen J. Newman[4]

[1] Joint Institute for Marine and Atmospheric Research, University of Hawaii, Honolulu, HI, USA
[2] Hawaii Institute of Marine Biology, University of Hawaii, Honolulu, HI, USA
[3] Department of Marine and Wildlife Resources, Pago Pago, AS, USA
[4] Department of Primary Industries and Regional Development, Western Australian Fisheries and Marine Research Laboratories, North Beach, WA, Australia

Corresponding author
Brett M. Taylor,
brett.taylor@noaa.gov

## ABSTRACT

The age-based life history of two commercially-important species of snapper (Lutjanidae) and one emperor (Lethrinidae) were characterized from the nearshore fishery of Tutuila, American Samoa. Examination of sagittal otoliths across multiple months and years confirmed the annual deposition of increments and highlighted marked variation in life-history patterns among the three meso-predator species. The humpback red snapper *Lutjanus gibbus* is a medium-bodied gonochoristic species which exhibits striking sexual dimorphism in length-at-age and consequent growth trajectories and has a life span estimated to be at least 27 years. The yellow-lined snapper *Lutjanus rufolineatus* is a small-bodied gonochore with weak sexual dimorphism, early maturation, and a short life span of at least 12 years. The yellow-lip emperor *Lethrinus xanthochilus* is a large-bodied species with a moderate life span (estimated to be at least 19 years in this study), rapid initial growth, and a more complex sexual ontogeny likely involving pre- or post-maturational sex change, although this remains unresolved at present. Ratios of natural to fishing mortality indicate a low level of prevailing exploitation for all three species, which is supported by low proportions of immature female length classes captured by the fishery. However, considerable demographic variability among the three species highlights the value of detailed age-based information as a necessary component for informing monitoring efforts and future management decisions.

# INTRODUCTION

Assemblages of snappers (Family Lutjanidae) and emperors (Family Lethrinidae) are highly valuable components of tropical insular fisheries across the Indo-Pacific region (*Carpenter & Allen, 1989*; *Dalzell, 1996*; *Newman, Williams & Russ, 1997*). Both families represent meso-predator assemblages that prey on small fishes and invertebrates and are targeted by commercial, recreational, and artisanal fishers throughout their range. The majority of species for both families are commercially valuable and are most often

harvested by line fishing techniques in depths from shallow reef-associated habitats (<10 m) to deep slopes extending to 100 s of meters (*Dalzell, Adams & Polunin, 1996*; *Newman et al., 2016*). However, in many insular locations throughout the tropics, snappers and emperors are also targeted in net and spear fisheries (*Dalzell, 1996*).

Life-history information underpins fisheries management in many nations. Age-based information is commonly derived from otolith-based studies, and the resultant age-based dynamics have been well-studied for both tropical snappers and emperors (*Heupel et al., 2010*; *Currey et al., 2013*). This age-based information represents a basis for guiding fisheries assessments and harvest strategies in many regions of the Indo-Pacific (*Newman et al., 2016*). Specific research topics include growth and mortality (*Newman, Williams & Russ, 1996a*; *Newman, 2002*; *Newman, Cappo & Williams, 2000a*, *2000b*; *Heupel et al., 2010*; *Currey et al., 2013*; *Ebisawa & Ozawa, 2009*), reproductive biology and spawning patterns (*Bannerot, Fox & Powers, 1987*; *Davis & West, 1993*; *Bean et al., 2003*; *Ebisawa, 2006*; *Grandcourt et al., 2010a*; *Heyman et al., 2005*; *Taylor & Mills, 2013*), and the spatial dynamics of age-based demographic variation (*Newman, Williams & Russ, 1996b*; *Kritzer, 2002*; *Williams et al., 2003*; *Taylor & McIlwain, 2010*). Harvested species representing the families Lutjanidae and Lethrinidae are known to have moderate to high longevity, ranging from 10 to 60 years, compared with other conspicuous reef fish taxa in tropical areas. In general, life span is positively related to body size across species for both families (*Heupel et al., 2010*; *Taylor, Oyafuso & Trianni, 2017*) and, therefore, vulnerability to overexploitation is often linked to body size.

Despite some basic commonalities between harvested snappers and emperors, the reproductive biology between these families is strikingly different. Snappers of the family Lutjanidae are, to date, unequivocally considered gonochoristic (*Newman et al., 2016*), whereby individuals are either male or female from earliest development and no sex change occurs throughout the life span. Ontogenetic reproductive development is much more variable across the emperors, with the dominant mode historically considered protogyny (female-to-male sex change after initial maturation as female; *Young & Martin, 1982*). However, functional gonochorism through juvenile hermaphroditism is now recognized in several species (*Ebisawa, 2006*; *Marriott et al., 2010*; *Taylor, Oyafuso & Trianni, 2017*). Both families are generally known to spawn in aggregations that are spatially predictable and synchronized with seasonal and tidal cues (*Johannes, 1978*, *1981*; *Taylor & Mills, 2013*). Characterization of the onset and timing of maturation and reproductive activity is most accurately achieved through histological examination of gonad material throughout the calendar year. This information is of prime importance to fishery managers as an essential input to region-specific stock assessments of exploited populations.

The U.S. Pacific Island Territory of American Samoa harvests nearly 300 fish species in their nearshore reef-associated fishery. Snappers and emperors presently comprise approximately 10% of the total catch by mass (*Pacific Islands Fisheries Science Center, 2018*). This fact, coupled with their comparatively high market value make them an important and highly desirable component of the fishery. The prevailing status of many reef-associated populations is poorly understood. The lack of appropriate catch trends

and life-history data for most targeted species highlights the need to evaluate the characteristics influencing a species' vulnerability to overexploitation. Therefore, the purpose of this study is to derive age-based life-history information from fishery-dependent collections of two snappers and one emperor of high commercial value to the nearshore, reef-associated fishery of Tutuila, American Samoa. The principal objectives are to estimate growth, life span, mortality, and reproductive parameters based on age estimates coupled with detailed length-sampling of the commercial fishery.

## MATERIALS AND METHODS

### Study area and sampling protocol

Commercial reef-associated fisheries were surveyed on the island of Tutuila, American Samoa (14.3 °S, 170.7 °W), from March 2011 to September 2015, through the NOAA Commercial Fisheries Biosampling Program (CFBP; *Sundberg et al., 2015*; IACUC Permit number 13-1696). Surveys were conducted from 5:30 to 8:00 am at roadside temporary vendors (typically individual or groups of fishers selling their catch from the previous night) or at the newly-established centralized fish market in Pago Pago. During sampling times, all landed fish were measured (nearest 0.1 cm fork length (FL)) and recorded by species. Subsamples of measured fish were purchased for life-history analysis for the following species: the yellow-lip emperor *Lethrinus xanthochilus*, the humpback red snapper *Lutjanus gibbus*, and the yellow-lined snapper *Lutjanus rufolineatus*. These three species are common commercial and subsistence targets in the American Samoa fishery, representing the 9th, 12th, and 66th most common species by mass, respectively, in the nearshore catch out of nearly 300 harvested species (*Pacific Islands Fisheries Science Center, 2018*). Purchased fish were selected non-randomly to encompass all length classes targeted by the fishery. For each purchased specimen, samplers measured length (nearest 0.1 cm FL) and total body mass (g). Sagittal otoliths and gonad lobes were surgically extracted from each specimen for age and reproductive assessment. Otoliths were cleaned with ethanol and stored dry in individually labeled vials. Gonads were weighed to the nearest 0.001 g and macroscopically designated by sex. Entire gonad lobes or cross sections (3 mm thick) of gonad material from the mid-sections of gonad lobes were removed and stored using individually-labeled histological cassettes in a 10% buffered formalin solution.

### Age determination and growth

For the three species, one sagittal otolith from each specimen was selected at random and affixed to a glass slide using thermoplastic glue (Crystalbond 509®), such that the primordium was focused just inside the edge of the slide, and the sulcul ridge was perpendicular to the slide edge. The otolith was ground to the slide edge using a 600-grit diamond lapping wheel with continuous water flow along the longitudinal axis until flush with the edge of the slide. The otolith was then removed with heat (~200 °C) and reaffixed with the newly flat surface down and ground to produce a thin (~250 μm) transverse section encompassing the core material. Annuli, represented by alternating translucent and opaque bands, were counted along a consistent axis on the face of the
sections to derive an estimate of age in years. Using reflected light (*L. xanthochilus*) and transmitted light (*L. gibbus* and *L. rufolineatus*) on a stereomicroscope, age readings for all specimens were conducted on three separate occasions by the primary author and final age (number of annuli as a proxy for age in years) was assigned when agreement in counts occurred. If three counts differed by one presumed annulus (e.g., 13, 12, 14), the middle age was assigned (e.g., 13). Daily growth increments were enumerated for one small (10.1 cm) specimen of *L. xanthochilus* with preparation and reading protocols following *Taylor & Choat (2014)*. Daily growth increment profiles of recently recruited *L. xanthochilus* were first summarized in *Wilson & McCormick (1999)*.

The assumption that annuli are deposited on a yearly cycle was tested using edge-type analysis for each species (*Manickchand-Heileman & Phillip, 2000*). The outermost otolith margin was scored as within either an opaque or translucent zone across all specimens, and proportions of opaque zone deposition were plotted across the calendar year for each species. Plots were presented alongside annual patterns of sea surface temperature, using remotely sensed data (one day resolution, Pathfinder data base available through NOAA Coastwatch, Jan 2006 through May 2011).

Sex-specific patterns of growth were modeled by fitting the von Bertalanffy growth function (VBGF) to length-at-age data using least squares estimation. The VBGF is represented by:

$$L_t = L_\infty \left[ 1 - e^{-K(t-t_0)} \right]$$

where $L_t$ represents the predicted mean FL (cm) at age $t$ (years), $L_\infty$ is the mean asymptotic FL, $K$ is the coefficient used to describe the curvature of fish growth toward $L_\infty$, and $t_0$ is the hypothetical age at which FL is equal to zero, as described by $K$. As the sampling protocol was fisheries-dependent, very few specimens were below the 15–20 cm range. Hence, to approximate early growth trajectories, VBGF models were constrained to an established or inferred (from similar taxa) length at settlement (i.e., FL at age 0). For *L. xanthochilus*, this was 3.0 cm (*Nakamura et al., 2010*); for *L. gibbus* and *L. rufolineatus*, this was 3.5 cm (*Mori, 1984*; *Nanami & Yamada, 2009*).

## Mortality

Total mortality ($Z$) was estimated using a multinomial catch curve fitted to the age classes at or above the assumed age at full recruitment to the fishery ($t_{rec}$). $t_{rec}$ is defined as one plus the peak frequency age (*Dunn, Francis & Doonan, 2002*). For fish at or above $t_{rec}$, the per recruit survival of fish ($S_t$) to integer age ($t$) was calculated as:

$$S_t = e^{-Z(t-t_{rec})}$$

$\hat{P}_t$, the expected proportion of fully-recruited fish at age $t$ was calculated as:

$$\hat{P}_t = \frac{S_t}{\sum_{t=t_{rec}}^{t_{max}} S_t}$$

where $t_{max}$ refers to the maximum observed age. The catch curve was fitted by maximizing the multinomial log-likelihood ($\lambda$) associated with the observed and expected proportions at age:

$$\lambda = \sum_{t=t_{rec}}^{t_{max}} f_t ln(\hat{P}_t)$$

where $f_t$ refers to the observed frequency at age $t$.

The value for the natural mortality (assumed constant across age) in this analysis was calculated using the *Hoenig (1983)* equation: $Z = e^{1.46 - 1.01 * log(t_{max})}$, whereby $Z$ is equal to $M$ through this derivation if the true maximum age was derived. For empirical derivations of $M$ in this study, we cautiously assumed this to be true.

Uncertainty in the estimate of total mortality was calculated using a length-stratified bootstrap procedure. The proportion in each length bin (5 cm bin width) from the full size distribution from the fishery was calculated and used as weights in the bootstrap resampling of the length-age samples. For example, if the sample size of the length-age data was 100 and the proportion of individuals in the first length bin in the full size distribution was 0.05, five data points from the first length bin in the length-age data were sampled with replacement. With the exception of a small individual in the *L. xanthochilus* dataset (10.1 cm, 0.35 years old (estimated based on daily growth increments)), the length ranges of the length-age data were similar to those from the full length distribution of the fishery. Each dataset was bootstrapped 10,000 times. The median of the bootstrap distribution was reported and the 2.5th and 97.5th percentiles of the bootstrap distribution of total mortality were reported as the 95% percentile confidence interval. The sensitivity of the size of the length bins on the total mortality estimate was evaluated using length bins of 2, 5, 10, and 20 cm, and also across individual years to examine potential effects of compounding multi-year data. The total mortality estimate using a conventional bootstrap was also evaluated as a part of the sensitivity analysis. This length-stratified bootstrap method was used because our sampling program was not designed to derive a random sample representative of the fishery, whereas the full market survey was designed for this purpose. Additionally, all otoliths and gonads that were originally sampled were not retained for final processing, which further precluded us from considering the sample to be representative.

## Reproduction

Fixed sections of gonadal tissue were histologically processed at the John A. Burns School of Medicine at the University of Hawaii. Sections were imbedded in paraffin wax, sectioned transversely at 6 μm, and stained on microscope slides with haematoxylin and eosin. Slides were viewed under dissecting and compound microscopes with transmitted light to determine sex and level of reproductive development following criteria and terminology from *Brown-Peterson et al. (2011)*. The female maturation schedule by length and age was modeled for each species. Specimens were scored by their maturity status (immature, 0: representing "Immature" and "Developing" phases from *Brown-Peterson et al. (2011)*; mature, 1: representing "Spawning capable," "Regressing,"
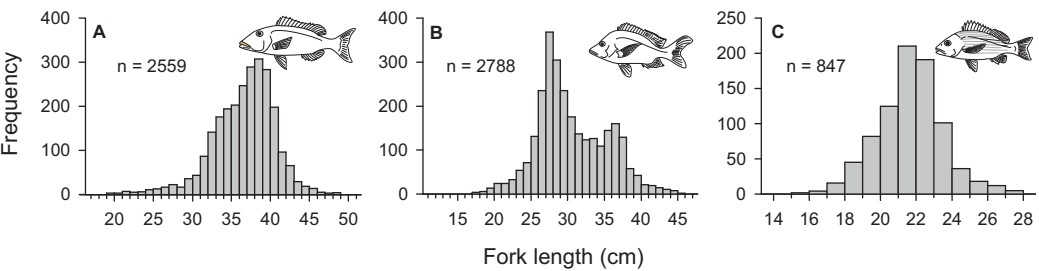

**Figure 1 Length-frequency distributions derived from the Tutuila-based, American Samoa commercial fishery from 2011 to 2015 for (A)** *L. xanthochilus,* **(B)** *L. gibbus,* **and (C)** *L. rufolineatus.* Total number of samples recorded are presented for each species.

and "Regenerating"), and maturity-at-length/age data were fitted with a two-parameter logistic curve as follows:

$$P_L = \left\{ 1 + e^{-\ln(19)(L-L_{50})/(L_{95}-L_{50})} \right\}^{-1}$$

where $P_L$ ($P_t$ for age at maturity) is the estimated proportion of mature females at a given length ($L$), and $L_{50}$ and $L_{95}$ ($t_{50}$ and $t_{95}$ for age) are the FL at 50 and 95% maturity, respectively. Corresponding 95% confidence limits (CLs) for the maturity schedules were derived by bootstrap resampling with replacement through 1,000 iterations.

Although not a primary objective of this study, we further examined aspects of the reproductive developmental ontogeny of *L. xanthochilus* based on histological features of specimens and length- and age-based patterns of sex ratio. Specifically, we searched for evidence of pre- or post-maturational female-to-male sex change, as has been commonly identified in species of *Lethrinus*, following criteria established in *Sadovy & Shapiro (1987)* and *Sadovy de Mitcheson & Liu (2008)*.

## RESULTS

### Age determination and growth

From March 2011 to September 2015, a total of 2,559, 2,788, and 847 commercial specimens were measured of *L. xanthochilus*, *L. gibbus*, and *L. rufolineatus*, respectively. Of these, 372 *L. xanthochilus*, 481 *L. gibbus*, and 217 *L. rufolineatus* were dissected for life-history analysis, but otoliths and/or gonads were retained for only 244, 311, and 149 individuals, respectively. Length-frequency distributions from the commercial harvest are displayed in Fig. 1. All species showed unimodal distributions with modal FL bins of 38.5, 27.5, and 21.5 cm, respectively. However, *L. gibbus* displayed a considerable second "hump" in the size distribution at 36.5 cm. Ages were estimated from sagittal otoliths from a total of 236 specimens for both *L. xanthochilus* and *L. gibbus* and from 134 specimens for *L. rufolineatus* (Table 1). Although no species were sampled through all calendar months, annual patterns of edge deposition demonstrated an annual periodicity in the formation of opaque zones, confirming that increments are indeed annual (Fig. 2; Supplemental Information). Opaque zones were fully deposited in August and September during peak low sea surface temperatures in American Samoa.

**Table 1 Summary of life-history trait estimates for three commercially harvested species from American Samoa.**

| | L. xanthochilus | | | L. gibbus | | | L. rufolineatus | | |
|---|---|---|---|---|---|---|---|---|---|
| | Males | Females | Combined | Males | Females | Combined | Males | Females | Combined |
| $L_\infty$ (cm) | 40.5 (39.7–41.3) | 39.6 (38.3–40.9) | 40.2 (39.5–40.9) | 38.8 (37.6–40.1) | 28.9 (28.4–29.4) | 32.9 (25.9–37.9) | 23.3 (22.7–23.9) | 21.8 (20.8–22.9) | 22.9 (22.4–23.5) |
| $K$ (year$^{-1}$) | 0.63 (0.56–0.72) | 0.67 (0.59–0.76) | 0.64 (0.59–0.69) | 0.32 (0.29–0.35) | 0.66 (0.58–0.79) | 0.46 (0.20–0.73) | 0.80 (0.69–0.96) | 0.86 (0.64–1.18) | 0.82 (0.70–0.99) |
| $t_0$ (year) | −0.12 | −0.12 | −0.12 | −0.29 | −0.19 | −0.25 | −0.20 | −0.20 | −0.20 |
| $n$ aged | 98 | 137 | 236 | 119 | 117 | 236 | 105 | 29 | 134 |
| $L_{50}$ (cm) | – | 30.0 (27.3–31.8) | – | – | 24.9 (23.5–25.8) | – | – | 16.4 (14.6–18.1) | – |
| $L_{95}$ (cm) | – | 39.9 (35.9–43.9) | – | – | 29.0 (27.5–30.4) | – | – | 23.5 (21.2–25.9) | – |
| $t_{50}$ (cm) | – | 2.1 (1.8–2.3) | – | – | 3.2 (2.9–3.9) | – | – | – | – |
| $t_{95}$ (cm) | – | 4.7 (4.1–5.1) | – | – | 4.8 (4.5–5.8) | – | – | – | – |
| $Z$ (year$^{-1}$) | – | – | 0.35 (0.29–0.41) | – | – | 0.22 (0.19–0.26) | – | – | 0.54 (0.44–0.75) |
| $M$ (year$^{-1}$) | – | – | 0.22 | – | – | 0.15 | – | – | 0.35 |

**Notes:**

Associated 95% confidence intervals are presented in parentheses where appropriate.

$L_\infty$, asymptotic length; $K$, growth coefficient; $t_0$, hypothetical age when length equals zero; $n$ aged, number of specimens used in age analysis; $L_{50}$, length at 50% sexual maturity; $L_{95}$, length at 95% sexual maturity; $t_{50}$, age at 50% sexual maturity; $t_{95}$, age at 95% sexual maturity; $Z$, instantaneous total mortality rate estimated from the logistic multinomial catch curve; $M$, natural mortality rate estimated from *Hoenig's* (1983) method.

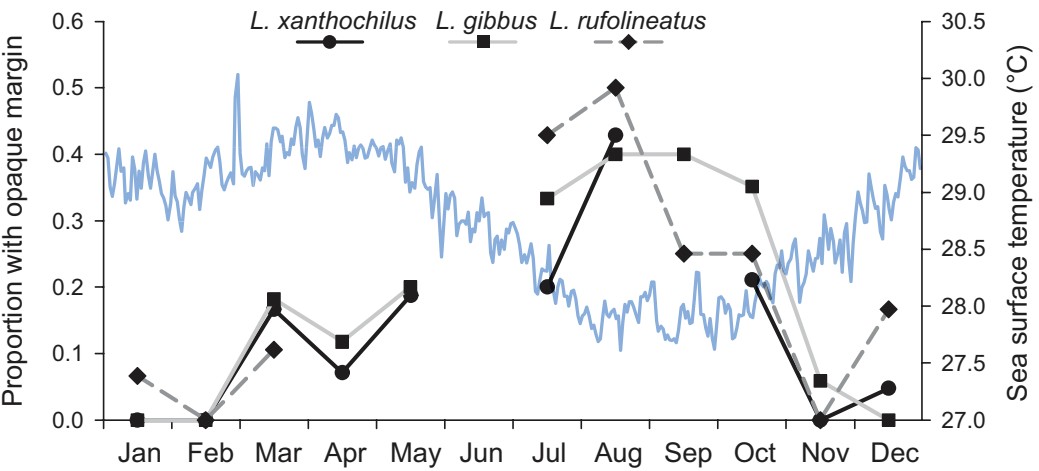

**Figure 2 Frequency of opaque edge deposition by month in transverse sections of otoliths from three commercial species from Tutuila, American Samoa.** The continuous blue line displays mean daily sea surface temperature across the calendar year based on satellite-derived data from Jan 2006 to May 2011.

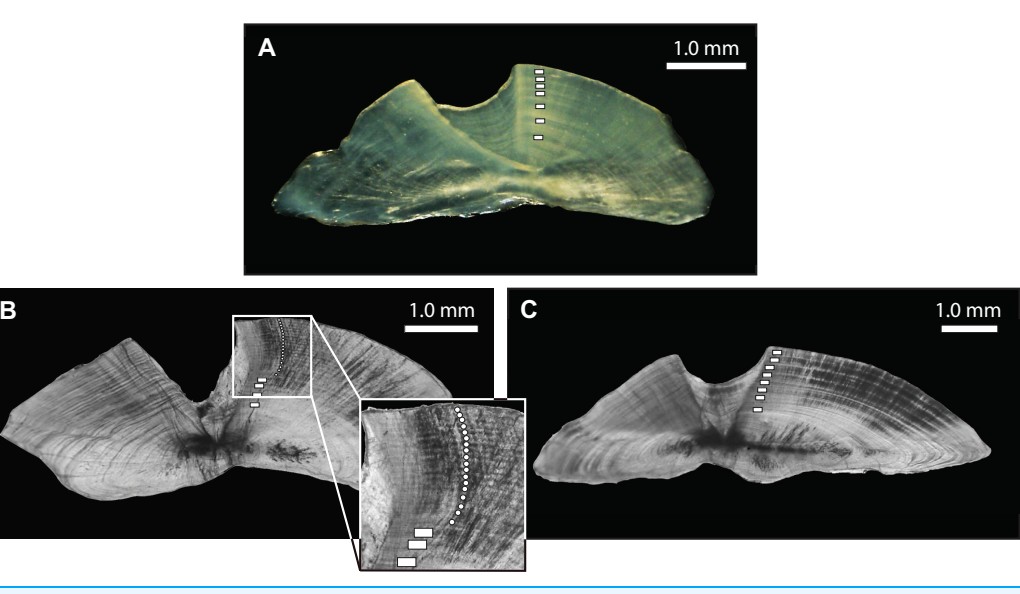

**Figure 3 Photomicrographs of transverse otolith sections for (A) *L. xanthochilus* (reflected light), (B) *L. gibbus* (transmitted light), and (C) *L. rufolineatus* (transmitted light).** Annual increments (opaque bands) are denoted by white boxes, with the inset in (B) providing greater resolution for smaller increments.

All three species deposited clearly defined annuli that were highly characteristic of otolith patterns previously identified for snappers or emperors (Figs. 3A–3C; *Marriott & Mapstone, 2006*; *Grandcourt et al., 2010b*). Otolith weight was a strong predictor for age in all species, whereby relationships were best explained using a standard quadratic equation (Fig. 4). The relationship between otolith weight and age did not differ between males and females for *L. xanthochilus* or *L. rufolineatus*, but it differed substantially between the sexes for *L. gibbus* (Fig. 4B). This pattern was retained in the length-at-age

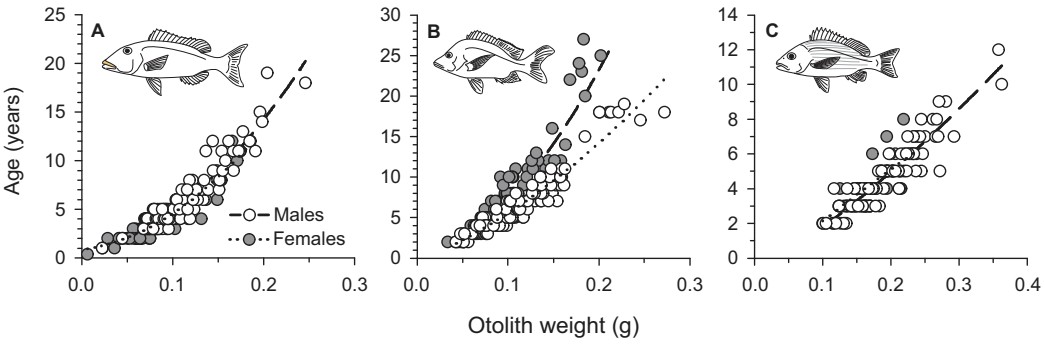

**Figure 4 Sex-specific relationships between sagittal otolith weight (g) and annual age (as number of annuli) for (A)** *L. xanthochilus,* **(B)** *L. gibbus,* **and (C)** *L. rufolineatus* **from Tutuila, American Samoa.** Equations are as follows: *L. xanthochilus*, male age = 0.3 + 20.2*(Otolith weight) + 247.3*(Otolith weight)$^2$, female age = 0.9 + 11.8*(Otolith weight) + 253.2*(Otolith weight)$^2$; *L. gibbus*, male age = 0.5 + 27.0*(Otolith weight) + 444.0*(Otolith weight)$^2$, female age = −0.5 + 47.4*(Otolith weight) + 130.3* (Otolith weight)$^2$; *L. rufolineatus*, male age = −1.0 + 25.8*(Otolith weight) + 21.1*(Otolith weight)$^2$, female age = −1.1 + 33.1*(Otolith weight) − 10.5*(Otolith weight)$^2$.

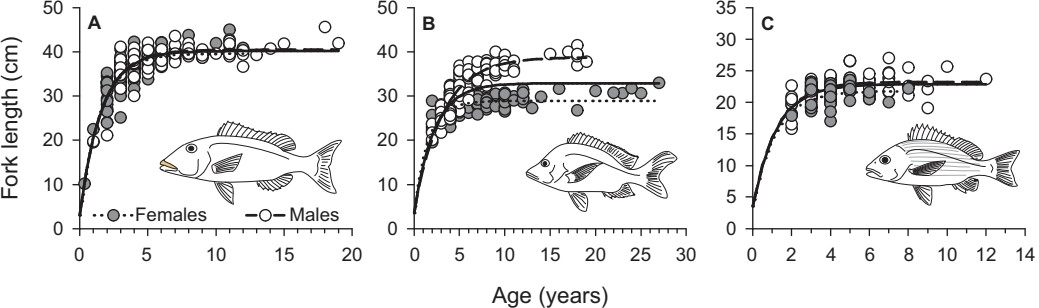

**Figure 5 Sex-specific and combined von Bertalanffy growth curves of (A)** *L. xanthochilus,* **(B)** *L. gibbus,* **and (C)** *L. rufolineatus* **from Tutuila, American Samoa.** Solid lines represent best-fit curves for both sexes combined for each species.

derived growth profiles, whereby patterns of growth were nearly identical between the sexes for *L. xanthochilus* and *L. rufolineatus*. Growth profiles diverged between males and females for *L. gibbus*, with males reaching a much larger asymptote (difference of ~10 cm; Fig. 5; values in Table 1). Overall VBGF parameter values $L_\infty$ and $K$ for the combined sexes were as follows: *L. xanthochilus*, $L_\infty$ = 40.2 cm, $K$ = 0.64 year$^{-1}$; *L. gibbus*, $L_\infty$ = 32.9 cm, $K$ = 0.46 year$^{-1}$; *L. rufolineatus*, $L_\infty$ = 22.9 cm, $K$ = 0.82 year$^{-1}$. Sex-specific VBGF values and associated CLs are presented in Table 1. Sex-specific maximum observed ages for each species were 12 and 19 years for female and male *L. xanthochilus*, respectively, 27 and 19 years for female and male *L. gibbus*, and eight and 12 years for female and male *L. rufolineatus*.

## Mortality
Median total mortality estimates were 0.35 (95% percentile CI [0.29–0.41]), 0.22 (95% percentile CI [0.19–0.26]), and 0.54 (95% percentile CI [0.44–0.75]) for *L. xanthochilus*, *L. gibbus*, and *L. rufolineatus* based on data pooled across surveyed years

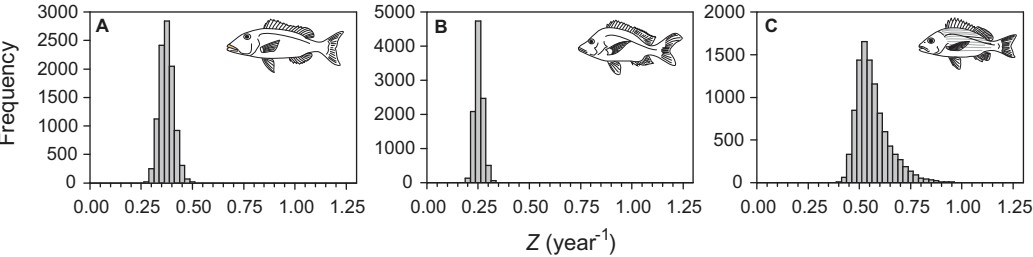

**Figure 6 Distributions of total mortality ($Z$, year$^{-1}$) from a length-stratified bootstrap resampling procedure followed by multinomial catch curve analysis.** Distributions of total mortality ($Z$, year$^{-1}$) from a length-stratified bootstrap resampling procedure followed by multinomial catch curve analysis for (A) *L. xanthochilus*, (B) *L. gibbus*, and (C) *L. rufolineatus* from the commercial fishery of Tutuila, American Samoa.

(Fig. 6). The estimate for total mortality and the distribution of total mortality bootstrapped estimates were sensitive to neither the size of the length bin, nor to the type of bootstrap procedure or timeframe (annual versus pooled) of survey data (Supplemental Information). Estimates from individual years were consistent and slightly higher than for pooled years for *L. xanthochilus* alone. The distributions of bootstrapped total mortality estimates were approximately symmetric for the *L. xanthochilus* and *L. gibbus* datasets, while the distribution of bootstrapped total mortality estimates for *L. rufolineatus* had a slight positive skew (Fig. 6C). Estimated natural mortality rates were 0.22, 0.15, and 0.35 year$^{-1}$ for *L. xanthochilus*, *L. gibbus*, and *L. rufolineatus*, suggesting fishing mortality rates were approximately half the presumed natural mortality across species.

## Reproduction

We confirmed sexual identity and characterized female maturation profiles using histological sections of gonads from 161 *L. xanthochilus* (87 females), 157 *L. gibbus* (93 females), and 100 *L. rufolineatus* (20 females). Our fisheries-dependent sampling program covered the length and age ranges over which female maturation occurred for *L. xanthochilus* and *L. gibbus*. Length at maturation (but not age) was modelled for *L. rufolineatus* based on only three immature individuals, and therefore, the maturation profile and $L_{50}$ estimate should be interpreted cautiously. The median length at female maturity ($L_{50}$) was estimated at 30.0 cm (27.3–31.8 cm 95% C.L.) for *L. xanthochilus*, 24.9 cm (23.8–25.8 cm 95% C.L.) for *L. gibbus*, and 16.4 cm (14.6–18.1 cm 95% C.L.) for *L. rufolineatus* (Figs. 7A–7C). Age at female maturity was only modelled for *L. xanthochilus* and *L. gibbus*. Median age at maturity values were 2.1 years (1.8–2.3 years 95% C.L.) for *L. xanthochilus* and 3.2 years (2.9–3.9 years 95% C.L.) for *L. gibbus* (Figs. 7D–7E).

Further examination of male *L. xanthochilus* found that testes were consistently characterized by a well-developed ovarian lumen, and the vast majority of testes (>80%) contained internal parasites. We found one specimen (39 cm FL) that contained an atretic vitellogenic oocyte in the presence of proliferating male material, indicating post-maturational sex change (functional protogyny). This specimen also had large gonad

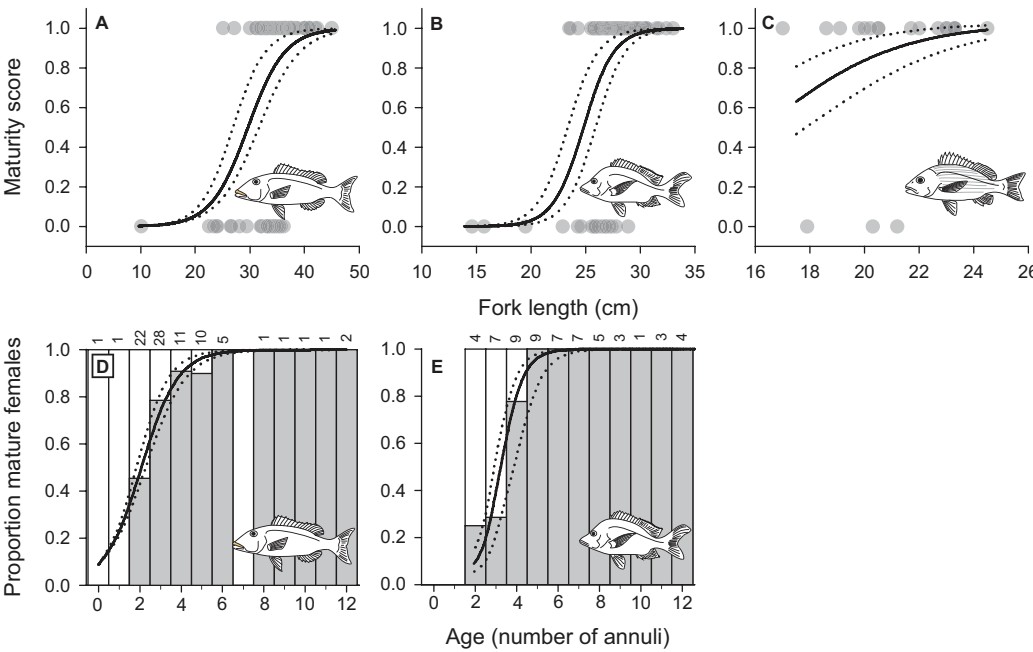

**Figure 7** Female maturation schedules by fork length fitted to maturity status data for (A) *L. xanthochilus,* (B) *L. gibbus,* and (C) *L. rufolineatus* from Tutuila, American Samoa. Age-based female maturation schedules were also derived for (D) *L. xanthochilus* and (E) *L. gibbus.* In (D and E), white bars represent the proportion of immature females, grey bars represent the proportion of mature females, and numbers above bars represent age-specific sample sizes. For all plots, solid lines represent best-fit models and dotted lines represent 95% confidence limits.

walls and displayed tight rounded folds of the epithelium (Fig. 8). Testes of other specimens were found to be in similar condition—presumed to be in a transitional mode—but did not contain identifiable atretic oocytes. Further, we found at least two potentially bisexual individuals (both aged two years) under or near the size at first female maturation (24.0 and 29.4 cm FL; Fig. 9), indicating primary development into males. In both cases, primary oocytes were sparse (compared with other immature females), and stromal tissue proliferated with tenuous evidence of sperm crypt development. Length-based and age-based relative proportions of females and males across length and age classes provided further evidence of both pre- and post-maturational sex change (Supplemental Information). It is clear that sex ratios change considerably across size and age classes and that some females are retained in the largest length classes, but more robust investigations are required to determine developmental ontogeny for this species.

## DISCUSSION

Compared with other harvested families in tropical fisheries globally, lutjanids, and lethrinids are among the most studied on coral reefs with regard to life-history information as a necessary input for fisheries management. The information presented in this study provides detailed demographic profiles for three commercially-important meso-predators that differ markedly in terms of their trait values and life-history patterns.

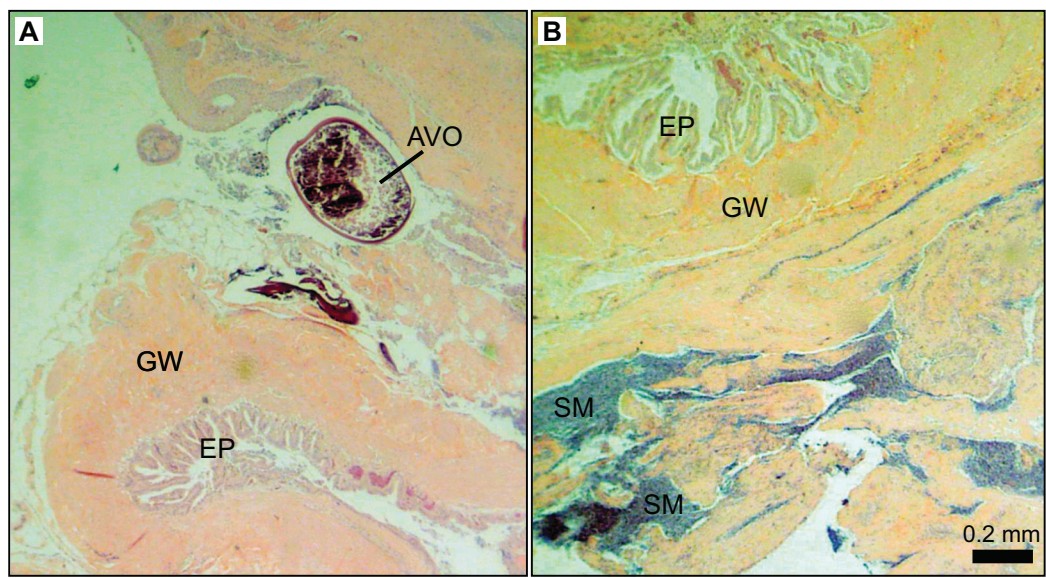

**Figure 8 Photomicrographs of a 39 cm fork length (six years old) *L. xanthochilus* with preliminary evidence of post-maturational female-to-male sex change.** (A) and (B) are from the same individual. Study sites: AVO, atretic vitellogenic oocyte; GW, gonad wall; EP, rounded folds of the epithelium; SM, spermatogenic material.

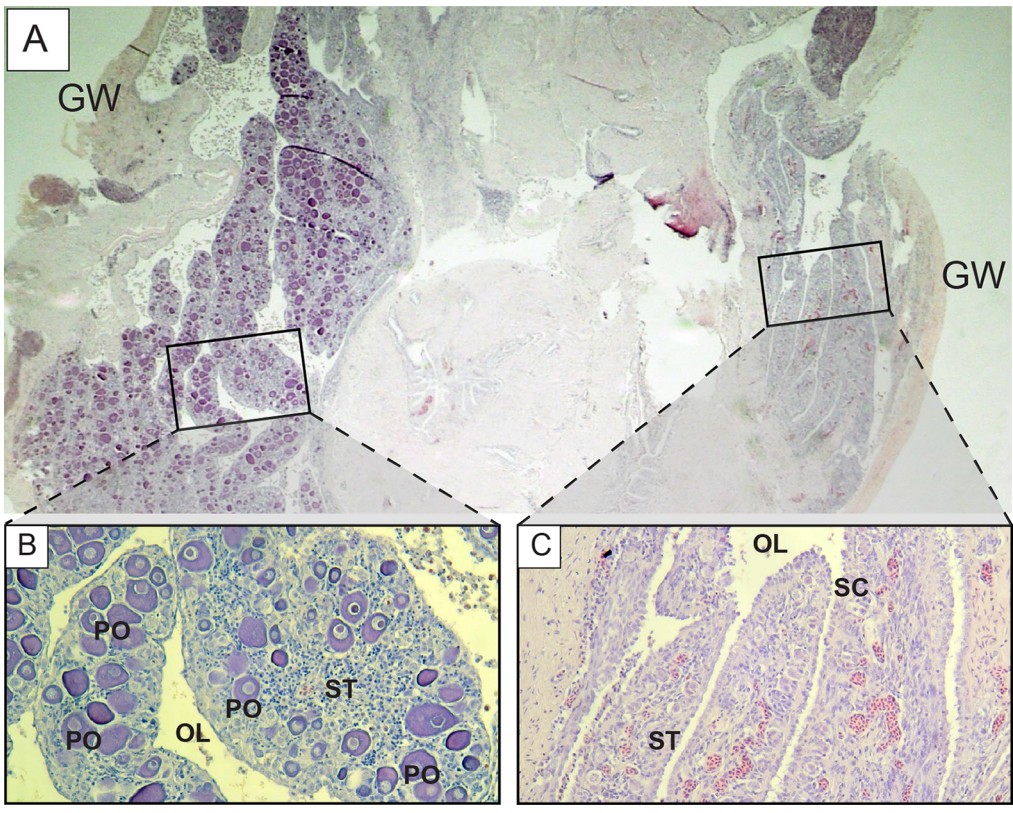

**Figure 9 Photomicrographs of a potentially bisexual *L. xanthochilus* (24 cm fork length, two years old).** (A) shows whole structure of gonad cross-section, while (B) and (C) provide greater resolution of characteristic features. GW, gonad wall; PO, primary oocyte; OL, ovarian lumen; ST, stromal tissue; SC, potential sperm crypt development.

*L. gibbus* is a comparatively medium-bodied gonochoristic species that has clear sexual dimorphism, and a long life span. *L. rufolineatus* is a small-bodied gonochore that has weak sexual dimorphism and a short life span. *L. xanthochilus* is a large-bodied species with a moderate life span and has a more complex sexual ontogeny likely involving pre- and post-maturational sex change. Hence, it is prudent to assume that responses to selective fishery harvest will differ among the three species. For instance, larger body size and longer life span are commonly associated with greater vulnerability to overexploitation (*Abesamis et al., 2014*). The influence of differing reproductive strategies on species' vulnerability is less clear.

This preliminary investigation found all three species to have low fishing mortality rates relative to the rate of natural mortality in American Samoa, with the vast majority of harvested fishery specimens greater than the median length at female maturation for each species. Estimated total mortality was derived using a length-stratified bootstrapping procedure which was robust to potential sampling bias for age-based data because it resampled age-at-length data to match the larger and unbiased market survey length distributions. The uncertainty surrounding estimates of total mortality decreased with increasing life span (i.e., number of data points yielding mortality curves) across species, but we caution the use of this method with *L. gibbus*, the longest lived species. This species demonstrated pronounced sexually dimorphic growth with males reaching a much larger length-at-age than females. Hence, the structure of length-frequency distributions is highly sex-specific, potentially yielding more variable age-frequency distributions when derived from age-length relationships based on pooled sexes. The data herein should be used in comprehensive stock assessments to fully explore the relative sustainability of extant fisheries. However, further simulation testing should first be conducted to understand the robustness of mortality estimates under different sampling designs and life-history traits using the length-stratified bootstrapping procedure.

To date, little age-based or reproductive information is available for the three species examined here. The most comprehensive description is for *L. gibbus* from Okinawa (*Nanami et al., 2010*), which documented life span, growth, and reproductive seasonality for the region based on fishery-dependent sampling. Sex-specific maximum ages and growth profiles from Okinawa were similar to those observed from American Samoa. *Nanami et al. (2010)* found that *L. gibbus* had a long (~6 month) spawning period which occurred during the warmer part of the year, whereby subsequent settlement of recruiting juveniles was synchronized with lunar cycles. *Johannes (1981)* documented from the accounts of fishers in Palau that *L. gibbus* spawns around full moon throughout the year. Considerable differences in the growth profile and maximum age of *L. gibbus* were identified from the Great Barrier Reef (GBR) (*Heupel et al., 2010*), although this study reported concerns regarding undersampling of both the smallest and largest (and presumably oldest) size classes. *Loubens (1980)* reported a maximum age estimate of only 18 years for *L. gibbus* in the waters of New Caledonia based on counts of annuli in sectioned otoliths, much less than the maximum age estimate derived from this study. For *L. rufolineatus*, *Mizenko (1984)* documented evidence for seasonal reproductive activity in Samoa during autumn and winter with spawning taking place around full moon. Similar
to our study, *Mizenko (1984)* found very few fish with immature ovaries above 17 cm, thus confirming our small estimate of length at maturity despite low sample sizes in the smallest length classes. We found no existing age-based or reproductive information for *L. xanthochilus* in the primary literature. Sample sizes of females across the annual and lunar calendar in the present study were not sufficient to determine reproductive periodicity. However, distinct post-ovulatory follicles were present for all species at various times during the year in mature females, suggesting that each species has an extended reproductive period.

Our results suggest a diandric protogynous life-history mode for *L. xanthochilus*. However, based on our relatively small sample size and limited evidence, we do not consider our results to be definitive regarding the reproductive ontogeny of this species in American Samoa. Most lethrinids are recognized as either functional protogynous hermaphrodites or juvenile hermaphrodites (*Ebisawa, 2006*). We recognized two potential pathways for male recruitment, pre-maturational and through sex transition from functionally mature females. Post-maturational male development was supported by length- and age-based schedules of sex ratio. The length-based schedule suggests that females occur into the largest length classes, and appears confounded by potential primary males (i.e., individuals that develop first as males). However, length-based schedules of sex change can be confounded by sexually-dimorphic growth, which was limited in *L. xanthochilus*. The age-based schedule demonstrates much clearer evidence of sexual transition from female to male as a cohort ages. Ultimately, length-based schedules of sex change are rarely clear cut for lethrinid species (*Ebisawa, 2006*) compared with prominent protogynes, such as labrids, that have dichromatism and seemingly much more complex social-reproductive systems centered on size-based defense of territories (*Taylor & Choat, 2014*).

Pronounced sexual dimorphism does not appear to be prevalent among the snappers, but considerable differences in asymptotic lengths between males and females have been observed for several species (*Newman, Cappo & Williams, 2000a*; *Newman, 2002*; *Shimose & Tachihara, 2005*). Here, differences in size-at-age between male and female *L. gibbus* matched observations previously documented from Japan for the species (*Nanami et al., 2010*) and represent the largest disparity in male versus female size in any snapper species to date. The ecological context of sexual dimorphism is hinged on assumptions that female size reflects a trade-off between growth and fecundity, and male size is influenced by reproductive competition (*Parker, 1992*). Based on the ubiquity of absent or weak sexual dimorphism in the snappers, we must presume that *L. gibbus* populations have an unusually different reproductive ecology compared with other lutjanids to yield such stark differences. Further, this level of dimorphism can complicate assessments of population status, especially when the primary data source is abundance and length-frequencies from fishery-dependent or independent surveys. We note that the frequency distribution presented in Fig. 1 for *L. gibbus* was bimodal, with primary and secondary peaks closely corresponding with asymptotic lengths of females and males. This highlights the importance of understanding how sex-specific growth trajectories influence sex ratios across length classes at the population level.

## CONCLUSION

Snapper and emperor species have historically been a highly-valued component of coral reef-associated fishes. Our results provide key population-level trait values that are of significant utility to stock assessment and fishery management. In doing so, we highlight differences in the biology and associated recovery potential of three phylogenetically-related species commonly harvested by the same methods during the same fishing trips.

## ACKNOWLEDGEMENTS

The authors thank the American Samoa Biosampling Team for their efforts in the field and R Nichols, E Reed, and K Rhodes for additional interpretations and discussions of gonad histology. Constructive comments by Tiffany Sih, Bill Duffy, and an anonymous reviewer greatly improved the manuscript. All data used in this publication are from https://inport.nmfs.noaa.gov/inport/item/5619.

### Funding

This study was funded by the National Marine Fisheries Service Biosampling Initiative and the Joint Institute for Marine and Atmospheric Research Territorial Biosampling Project 6105137. There was no additional external funding received for this study. The funders had no role in study design, data collection and analysis, decision to publish, or preparation of the manuscript.

### Grant Disclosures

The following grant information was disclosed by the authors:
National Marine Fisheries Service Biosampling Initiative and the Joint Institute for Marine and Atmospheric Research Territorial Biosampling Project 6105137.

### Competing Interests

The authors declare that they have no competing interests.

### Author Contributions

- Brett M. Taylor conceived and designed the experiments, performed the experiments, analyzed the data, contributed reagents/materials/analysis tools, prepared figures and/or tables, authored or reviewed drafts of the paper, approved the final draft.
- Zack S. Oyafuso performed the experiments, analyzed the data, contributed reagents/materials/analysis tools, authored or reviewed drafts of the paper, approved the final draft.
- Cassandra B. Pardee performed the experiments, analyzed the data, contributed reagents/materials/analysis tools, authored or reviewed drafts of the paper, approved the final draft.

# PeerJ

- Domingo Ochavillo conceived and designed the experiments, performed the experiments, contributed reagents/materials/analysis tools, authored or reviewed drafts of the paper, approved the final draft.
- Stephen J. Newman analyzed the data, contributed reagents/materials/analysis tools, authored or reviewed drafts of the paper, approved the final draft.

## Animal Ethics

The following information was supplied relating to ethical approvals (i.e., approving body and any reference numbers):

All research was carried out under permit number 13-1696 issued by the University of Hawaii Institutional Animal Care and Use Committee.

## Data Availability

All raw data used for figures and analyses in the manuscript are supplied as Supplementary Information. All data used in this publication are from *Pacific Islands Fisheries Science Center (2018)*: American Samoa Commercial Fisheries BioSampling (CFBS), https://inport.nmfs.noaa.gov/inport/item/5619. The provided data are de-identified from an otherwise confidential database and the authors have permission to share this data here.

## Supplemental Information

Supplemental information for this article can be found online at http://dx.doi.org/10.7717/peerj.5069#supplemental-information.

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
