# Peer review of "Comparative demography of commercially-harvested snappers and an emperor from American Samoa"

_PeerJ, doi:10.7717/peerj.5069_

## Round 0.1 · original submission · Minor Revisions

Your manuscript received 3 expert reviews, who are unanimous regarding the value of the study, and were generally impressed with both the quality of the work and the writing. They provide some feedback for improvements to the work that I expect will be simple to incorporate and will make an already good paper even better.

The one additional question you may wish to address in revision is that it is unclear how exactly you calculated annual mortality from the 5 year spread in the data used to estimate Z. Some might worry that adding samples over 5 years could allow for aging of the cohort resulting in greater numbers of larger, older individuals than would be captured in any given year alone, effectively resulting in an underestimation of total annual Z. To avoid criticism by those who might feel this is an issue, it might be good to just clarify your analytical approach here, or compare results of the total Z to the most-complete sampling year(s?) to indicate whether this concern is misguided or not.

Overall, I see nothing in the reviews that is likely to be an issue for you to address, and I do not expect it will require additional review. I look forward to seeing your revised manuscript.

Reviewer 1 ·

Basic reporting

Overall, I found this paper to be well-written, clear and interesting to read. The introduction was well structured to provide context, and both the introduction and discussion generally well referenced (though see one minor comment in General comments below).
The figures are relevant, and generally well captioned. The authors should, however, take care to ensure captions can be interpreted without referring back to the text (i.e. there should be no acronyms or abbreviations).
Raw data has been supplied in an easily interpretable form.

Experimental design

The paper generally uses well-established and repeated laboratory and analytical techniques. The research questions are generally well-defined and relevant to the management of these species in American Samoa. The paper falls within the editorial scope of PeerJ.

One concern I had with the experimental design is that the authors have not sufficiently explained how the sub-sample of each species used for life-history analysis was obtained from those collected for life-history analysis. For each species, there is a large discrepancy in the number of fish collected for life history analysis and that used for various ageing and reproductive (histological) analyses. For example, from the MS Excel data file provided there were 481 individual Lutjanus gibbus obtained for life history analysis, but only 157 were used for gonad histology and less than 50% (236) were aged. Similarly, of the 217 Lutjanus rufolineatus obtained for life history analysis, only 100 were used for gonad histology and only 134 were aged. I can appreciate that for gonad histology at least the intent was to examine female maturation profiles and thus there would be a discrepancy resulting from the exclusion of male fish, however there appear to be large numbers of females excluded as well. The authors make no statement of how or why the sub-sample obtained for life history analysis was further sub-sampled (although they do caution the validity of their findings given low sample sizes for maturation in L. rufolineatus). I found this a little misleading as I was at first quite impressed with sample sizes used in the study, and would thus like to see clarification from the authors.

Validity of the findings

Aside from those points raised above, I found the data to be generally robust and statistically sound. Conclusions are generally well stated and limited to supporting results (through see general comments below).

Additional comments

General comments
Line 18: what is the difference between life-history biology and reproductive biology? To me, reproductive biology is a component of life history, thus its inclusion in this opening sentence seems somewhat redundant.

Lines 31-32: suggest adding ‘female’ here (i.e. low proportions of immature female…’, as male maturity was not assessed.

Line 80: Reference for Commercial Fisheries Biosampling Program, 2018, is not included in reference list. Perhaps this should be Sundberg et al. 2015?

Line 208: Sample sizes stated here seem incorrect, as least compare to the raw data file. According to the raw data file, there were 372 Lethrinus xanthochilus, 481 Lutjanus gibbus, and 217 L. rufolineatus obtained for life history analysis?

Lines 290-286: I found the inclusion of information on schooling behaviour of each species in this section a little misleading, as the sentence that leads up to this sets the reader up for a summary of ‘the information presented in this study’. This study did not explore schooling behaviour, and I suggest the statements on schooling be referenced or, better yet, omitted.

·

Basic reporting

a. Clear and unambiguous, professional English used throughout.

• The writing is clear and professional.
• This is a very minor point and stylistic, but there are some instances where I questioned the word choice. I refer specifically to the use of ‘adequately’ (Line 250) and ‘tenuously suggest’ (Line 331), both instances have the connotation that they are pre-addressing reviewer concerns, but I question why this sub-text is necessary.

b. Literature references, sufficient field background/context provided.

• Missing reference Line 104-106, CFBP Report?
• Line 129: Any daily ring validation in similar species?
• Line 131: I would include a standard citation here on this technique.

c. Professional article structure, figs, tables. Raw data shared.

• Fig 2A: possible for temperature to be in light blue color or darker gray for ease of reading? Not distinctive from species lines/patterns and vision-impaired readers might have trouble even distinguishing the lines from the background color.
• Figure 3A and C: The female line is hard to see, possible to switch the pattern of male and female or overlay the lines on top of the points?
• I can understand the justification for including the gonad images (S1 and S2) in the Supplemental materials as they are not of ‘primary importance’ to your results. However, I would argue that they are relevant to your discussion, and therefore should be placed within the main part of the manuscript as SI often is overlooked. I have not found in the peer reviewer guidelines or the author instructions limitations on the figure/table numbers that would preclude you incorporating them in the main body of the manuscript. Further, they are of decent image quality and add to the Discussion.

• Raw data is included in Supplementary data


d. Self-contained with relevant results to hypotheses.

The authors have contained their findings to the hypotheses.

Experimental design

a. Original primary research within Aims and Scope of the journal.

This primary research falls within the Aims and Scope of PeerJ.

b. Research question well defined, relevant & meaningful. It is stated how research fills an identified knowledge gap.

While this research might be described as basic demographic research, the authors have presented an interesting comparison among the three species in a way that is meaningful for multispecies fisheries management.

c. Rigorous investigation performed to a high technical & ethical standard.

Study was performed to a high technical and ethical standard typical of fisheries-dependent studies. There is a statement of animal ethics approval.

d. Methods described with sufficient detail & information to replicate.

Methods were detailed and easy to follow.

Validity of the findings

a. Impact and novelty not assessed. Negative/inconclusive results accepted. Meaningful replication encouraged where rationale & benefit to literature is clearly stated.

The authors argue their findings persuasively, but have acknowledged where there is need for greater replication or weaker results. Examples of this are easy to find in the manuscript:

Line 251: the authors anticipated that this was weaker result, but they have specifically stated that it should be interpreted with caution.
Line 331: “Our results tenuously suggest… small sample size…etc”

b. Data is robust, statistically sound, & controlled.

The authors have provided data and methods are mostly standard for the field.

c. Conclusion are well stated, linked to original research question & limited to supporting results.

Yes

d. Speculation is welcome, but should be identified as such.

Some ‘speculation’ feels inadequately discussed. For example, Line 288 “responses…will differ among the three species”. I wish the authors would have expanded more on that thought or written more clearly their intention – at present it reads as incomplete and perhaps intentionally vague.

Additional comments

This manuscript is clear, the results and methods are sound, and data are elegantly presented. Demographic information like age and growth information is of utmost importance for fisheries management. The authors have written a compelling argument for contrasting life-history patterns among lutjanid/lethrinids, provided detailed methods and raw data, clearly stated where their results should be interpreted with caution, and discussed some interesting aspects of reproductive ecology. I see no reason why this research should not be published in PeerJ.

·

Basic reporting

The manuscript is well written and thoughtfully organized. There is enough background information and relevant citations. For the most part the tables and figures are well done, I have a couple of issues will I will address below. In the raw data I would have liked to see the agees for all three ageing attempts and not just a final age. I will have more to say about that in another section. Overall it was a very enlightening paper and adds a lot of new information to the existing literature.

Experimental design

The research question is well defined and the methods used to answer the questions were appropriate. I have a few comments, first was there more than one person who aged the fish? Did one person age teh fish 3 times? Either way, I would have loved to see each persons age for each fish. In any ageing study, precision estimates are very helpful indeterminate inf the structure being used to age is the appropriate one. I also didn't see how the authors came up with the equations for the relationship between sagittal otolith weight and annual age. in the results it was state that "Otolith weight was a strong predictor for age in all species, whereby the relationships were best explained using a standard quadratic equations fig.3" I would have liked to see how you came up with those numbers. Also, Otolith weight and fish length could also be strongly correlated, I am not questioning the results, I would just suggest explaining how you came up with the specific equations.

Validity of the findings

For the most part the findings and conclusions are supported by the data. The correct statistics and procedures were used to reach the conclusions. There is a lot of new information that will greatly add to the literature about the biology of these three species. However, some of the statements in the discussion and conclusions about the impact to the fishery need to be addressed. On line 290: All three species were found to have low fishing mortality rates ...." This is the first time fishing mortality was brought up. The statement is disingenuous, there is nothing in the manuscript that supports that statement.
Lines "301 - 303 Nevertheless, this preliminary information on mortality ratios and catch distribution relative to maturation size implies that these species are not approaching overexploitation in American Samoa" Once again, there is no talk about exploitation in a fishery anywhere in the manuscript. Other than a section in the introduction Line 82: that talks about "the status of many reef associate populations is poorly understood" This paper helps to address some of the data gaps to better understand the biology and some of the fishery characteristics, however until an assessment is conducted using the new information, the statements about exploitation and fishing mortality are a bit premature. Also in the conclusions, Line 369: "...associated recovery potential...," I didn't see anything in the manuscript that supports that statement.
Finally, in the introduction line 89: ..based on validated age estimates..were you trying to used the edge analysis to try to validate the ageing method, or that confirming that the annuls was formed once per year? I have a few specific comments about that below. Because you didn't address that in the discussion, I would alter the statement and not use the phrase "validated age", at most the results suggest a consensus age, which is something that would be really good to see in the raw data.

Additional comments

Overall, it was a very good manuscript. I have a couple of minor comments here.first: line 116: "...one sagittal otolith from each species was selected at random.." Why was that? most age and growth studies pick the one for consistency. line 214: Annual patterns of edge deposition clearly demonstrate... Figure 2A is very busy, it isn't clear at all. everything is overlaid and I had to expand the graph a lot to see what was going on. I would have figure 2a as a stand alone figure, I would also suggest a table with average edged deposition values per month, mainly because you don't have samples for all 12 months. Finally, line 425 & 427, is it RE or RL, Johannes?

---

## Round 0.2 · accepted · Accept

Thanks for your detailed responses to the minor concerns raised by the referees on your submission. The unanimous support of the 3 referees on the initial submission and minor suggestions raised in review preclude the need for additional review, and I am satisfied that you have addressed all concerns with your revision. I am happy to move your submission along to production.

#